# Phytochemistry, Biological and Pharmacological Activities of the *Anacyclus pyrethrum* (L.) Lag: A Systematic Review

**DOI:** 10.3390/plants11192578

**Published:** 2022-09-30

**Authors:** Hanane Elazzouzi, Kamal Fadili, Ali Cherrat, Smail Amalich, Nadia Zekri, Hannou Zerkani, Imane Tagnaout, Christophe Hano, Jose M. Lorenzo, Touria Zair

**Affiliations:** 1Research Team of Chemistry, Bioactive Molecules and Environment, Laboratoire des Matériaux Innovants and Biotechnologie of Naturelles Ressources, University Moulay Ismail Faculty of Sciences, Zitoune, Meknès B.P. 11201, Morocco; 2Laboratory of Spectroscopy, Molecular Modeling, Materials, Nanomaterials, Water and Environment, University Mohammed V Faculty of Sciences, 4-Avenue IbnBattouta, Rabat B.P. 1014 RP, Morocco; 3Laboratory of Phytochemistry, National Agency of Medicinal and Aromatic Plants, Taounate 34000, Morocco; 4Laboratoire de Biologie des Ligneux et des Grandes Cultures, INRA USC1328, Orleans University, CEDEX 2, 45067 Orléans, France; 5Centro Tecnológico de la Carne de Galicia, Rúa Galicia No 4, Parque Tecnológico de Galicia, San Cibraodas Viñas, 32900 Ourense, Spain; 6Área de Tecnología de los Alimentos, Facultad de Ciencias de Ourense, Universidad de Vigo, 32004 Ourense, Spain

**Keywords:** *Anacyclus pyrethrum*, medicinal plants, traditional medicine, phytochemical, biological activity

## Abstract

*Anacyclus pyrethrum* (L.) (Asteraceae) is an important annual medicinal herb and is widespread in Morocco and Algeria. Most of its parts are used in traditional medicine and the roots are the most important parts used. The present review gives an account of the updated information on its phytochemical and pharmacological properties. We have collected the essential characteristics and the different scientific data of the *A. pyrethrum* species, and reviewed its potential. It is seen from the literature that *A. pyrethrum* is a rich source of the phytochemical constituents such as alkaloids (pellitorin) and *n*-alkylamides. This species also contains pyrethrins, sesamin, traces of essential oils and a wide range of other chemical compounds. These active substances possess antimicrobial and anti-inflammatory activities. The plant has an antidiabetic, insecticidal and immunostimulatory effect, as well as an aphrodisiac and antioxidant potentials, and various other important medicinal properties. Many traditional uses are also reported in previous research such as for rheumatism, sciatica, colds, neuralgia and paralysis. This species is considered to be a sialagogue, and used in the treatment of stomach ailments, diseases of inflammation of the mouth, against cysts in the genital tract and to relieve toothaches. Thus, further research must be carried out in order to establish any relationship between the traditional uses, phytochemistry and toxicity. Moreover, *A. pyrethrum* is quite promising as a medicinal agent, so further clinical trials should be performed to prove its efficacy.

## 1. Introduction

Due to their effectiveness with low side effects, medicinal plants have an important role in food systems, cosmetic use and are commonly used for the cures and prevention of some diseases. All over the world, the traditional use of medicinal plants as a natural remedy for various diseases has received a great amount of attention from the scientific community and it has occupied an important place in medicine. Medicinal plants are a very valuable source for the production of new miraculous chemical molecules that are in great demand in the pharmaceutical, food, cosmetic and perfume industries [1].

These molecules are often likened to active ingredients with specific properties that give them a unique character [2]. In fact, the aromatic and medicinal plants currently constitute an essential product in modern and traditional medicine. It has been reported that approximately 30% of the drugs prescribed by doctors are of natural origin, while 50% of those are sold without a prescription [1]. Indeed, the exploitation of this heritage suffers from a lack of accurate knowledge for the potential of the plant mass and chemical nature of its extracts. Therefore, it is necessary to perform scientific studies to verify the accuracy of the information transcribed through the ethno-botanical writings, to eliminate any baseless or even dangerous uses, and to rationalize the use of medicinal plants.

*A. pyrethrum* (L.) is one of the spontaneous plant species of the Asteraceae family, endemic to Morocco and Algeria [3,4]. This species includes the two varieties, the *A. pyrethrum* var. *pyrethrum* (L) and the *A. pyrethrum* var. *depressus* (Ball) Maire [5,6]. Botanical and systematic descriptions of this species have been discussed by several taxonomists in various flora publications [6]. Over the past few years, there has been an increase in the interest of research on the chemistry and biological activities of the *A. prethrum* species.

Several studies on the chemical composition of *A. pyrethrum* have been previously published [7,8], confirming the presence of a wide variety of phytochemicals, of which about one hundred different compounds have been described to date. In traditional medicine, the roots of *A. pyrethrum* are recommended for treating toothaches, salivary secretions, angina, digestive problems, lethargy, female infertility and even paralysis of the tongue and limbs. They are used in the form of cream-based animal fats to treat gout and sciatica and to keep illness away [6]. As reported in earlier data, the roots of *A. pyrethrum*. possess interesting pharmacological properties including anticancer, aphrodisiac, anticonvulsive, androgenic and fertilizing, antiparasitic and antibiotic, bioinsecticide, antidiabetic, antifungal and immunostimulant effects [5]. In recent years, we have seen the publication of review articles on this species. However, these articles only present the existing, broad knowledge of its therapeutic values in a very general way.

This study was performed by collecting all of the information and developing a database on the species through a comprehensive literature review, and consequently on the sustainability of our knowledge of *A. pyrethrum*. In fact, while compiling this review, every effort was made to present in detail the qualities of this species, with particular emphasis on the current work, on the chemistry of the plant, on the chemistry of its essential oils, its variability in chemical composition in order to evaluate the importance of this species in the traditional phytotherapy, as well as in modern medicine. The emphasis was put on the latest biological activities confirmed by scientific studies. Moreover, the studies on the safety of its use were reported. Therefore, more studies are needed to confirm its efficacy and toxicity with research-based evidence, as plant-derived agents may be cost-effective and safer alternatives. The funds should be made available to educational institutions to conduct research on various other animals for future clinical trials and validate the safer use of its herbal derivatives in the biological process.

## 2. Botanical Aspects

### 2.1. Morphology

*A. pyrethrum* is a perennial plant belonging to the Asteraceae family. It is a species with a height of 40 to 60 cm and is characterized by numerous simple or small branching stems growing from the ground, and bearing leaves that are finely cut, delicate and pubescent. Its yellow-hearted flowers consist of white ray florets on the inside and purple on the outside. The roots are long, thick, fibrous, brown on the outside and white on the inside (Figure 1). All fruits (Achene) are bald or with a faint crown [4,9,10]. The species blooms between May and August [11].

The ethnological surveys conducted in Morocco, on the plant show that it is highly regarded by the local population for its medicinal properties and especially for its social and economic values. Thus, from a therapeutic point of view, the root is the most important and most widely used part of the plant [1,12].

### 2.2. Taxonomy and the Geographical Distribution

*A*. *pyrethrum* is a Moroccan and Algerian endemic species of the Asteraceae family [3]. Some authors state that it is endemic to Morocco, Algeria and India [9], and according to Boulos (1983), *A. pyrethrum* is mainly distributed in North Africa, the Mediterranean region and India [13,14]. It is shown in another study that this species is widespread in North Africa, Central Asia and the Xinjiang region of China [15,16]. The common name of *A. pyrethrum* is African *pyrethrum*. The names in Arabic, Moroccan and North African colloquial languages are: Awd al-Attas, Akkar Karra, Tighuendeste or Ighuendes Hallala, Arq-Echlouh, Aqirqarha, Kûkû [17]. The roots of *A. pyrethrum* were used in traditional medicine [18].

Several authors have reported that there are two types of *A. pyrethrum*: *Anacyclus pyrethrum* var. *pyrethrum* (L) and *Anacyclus pyrethrum* var. *depressus* (Ball) Maire. Furthermore, other authors have confirmed this result through an ethnobotanical study in the Azilal region of Morocco [1]. It is found that the two cultivars *A. pyrethrum* var. *pyrethrum*, locally known as “Ighuendez”, and *A. pyrethrum* var. *depressus*, which is known as “Tiguendizt” have distinct characteristics that differentiate them, such as the thickness of the collar, the wings of the fruit and the diameter of the head. In addition, the cultivar *A. pyrethrum* is characterized by roots that generally have a bitter taste with a narcotic effect and the basal leaves are dense and green with bright red petals on their undersides [11,13,19].

In Morocco, this species is found in cut forests, matorrals, hermes, grasslands, low plains, middle and high mountains, in cold semi-arid, semi-humid, humid, high mountains, at the level of the Saharan Atlas, against the Atlas, the High Atlas, Middle Atlas, North Atlantic Morocco, the plateaus of eastern Morocco and the Rif at altitudes between 1000 and 2500 m. The *A. pyrethrum* appears to be best adapted to well-drained soils and unshaded areas covered with trees or shrubs on the levels of vegetation over the Mediterranean [9]. The sexual reproduction is the most widespread (seed). The asexual reproduction (part of the plant or rhizome) is also possible. The modes of seed dispersal are through gravity, wind, water, animals (mammals, birds, insects, etc.) and humans [10].

### 2.3. Traditional Use

The *A. pyrethrum* is a plant widely used in traditional medicine for various diseases (Table 1). A study found that a decoction of the roots of *A. pyrethrum* is used in the treatment of stomach diseases and stomatitis [20]. In addition, the traditional use of the stalk powder from the *A. pyrethrum* mixed in honey, for cysts in the reproductive system, is very common in the population of Moulouya, in upper Morocco [21].

In 2014, an ethnobotanical study on *A. pyrethrum* was performed in the district of Meknes, El Hajeb, Khenifra, Azrou and Ifrane-Morocco [22]. It showed that the surveyed population uses *A. pyrethrum* to treat several diseases such as rheumatic diseases, gastrointestinal diseases, oral diseases, respiratory diseases, genitourinary diseases and skin diseases. In the ointment (in olive oil), the inhabitants of Khenifra, Morocco use the roots of *A. pyrethrum* in the treatment of abscesses, rheumatism and dermatitis [23].

In addition, an ethnobotanical study conducted in the Meknes-Tafilalet region (Morocco) on Asteraceae, showed that the surveyed population have used a decoction and the powder of the roots of *A. pyrethrum* to treat osteoarthritis disorders, stomatitis and inflammation of the urinary tract and genital organs [24]. Another ethnobotanical survey showed that the *A. pyrethrum* root powder is used to treat respiratory diseases [25]. In [26], an ethnobotanical study of plants exploited by cooperatives and associations in the Meknes-Tafilalet region of Morocco, showed that the infusion and decoction of the *A. pyrethrum* roots have been used for sore throats, toothaches and skin revitalization.

The ethnobotanical, ethno-taxonomic and ethnoecological study performed by Ouarghidi and Abbad, in the Aït Mhamed valley (Azilal region, Morocco) shows that the preparation most used in this region is turning the dry roots into powder and mixing the powder with honey. In addition, for the treatment of articular rheumatism, dental pain, intestinal pain and colic, the residents of this region chew the fresh roots of *A. pyrethrum* or use a decoction of these roots [1].

## 3. Phytochemistry of *Anacyclus pyrethrum*

### 3.1. Chemical Compounds of the Essential Oils

In the literature, the essential oils (EOs) have been extracted from *A. pyrethrum* via hydrodistillation, for a long time. This species is not rich in EOs, and its yield does not exceed 1%. Moreover, its EO production reaches its maximum during the flowering period. In Ben Slimane, in Morocco, this species yields 0.09% of EOs [7]. In addition, another study found that the EO yield of the spontaneous species collected in Timahdite (Middle Atlas) after flowering (0.07%) is relatively higher than that collected before flowering (0.05%) [27]. In Algeria, the species’ EO yield during the vegetative, flowering and post-flowering stages of growth are 0.011, 0.015 and 0.019%, respectively [28]. The observed intraspecific differences in the yields can be attributed to the harvest period.

The EO analyses of *A. pyrethrum* collected from the Ben Slimane region showed the presence of 32 compounds representing about 92.67% of the total chemical composition. The oxidative sesquiterpenes constitute the abundant group of all identified compounds (58.96%), followed by hydrocarbon sesquiterpines (24.19%). The main compounds are: spathulenol (20.47%), germacrene-D (16.48%), caryophylleneoxide (13.20%), 4-(14)-salvial-1-one (8.27%), caryophyllene-4(14) and 8(15)-diene5α-ol (7.30%) (Figure 2) [7]. The study of *A. pyrethrum* from Timahdit, Morocco, revealed the presence of 42 compounds for the sample collected before flowering and 36 compounds for the sample collected after flowering. These compounds represent 91.32% and 91.82%, respectively, of the total of these essential oils. The oxygenated sesquiterpenes constitute the most abundant group among the identified compounds. Its rate ranges from 89.17% (before flowering) to 90.58% (after flowering), during the maturity stage [27]. Similarly, this group is the most abundant Algerian species, as demonstrated in [28]. In this study, the proportion of sesquiterpenes ranged between 37.1% and 58.6% before and after the flowering, respectively [28].

In a study cited in [7], the percentage of the main component spathulenol increased significantly in the sample collected after flowering (16.9%), than that collected before flowering (13.31%). Thus, the proportion of germacra 4(15),5,10(14)-tren-1-α-ol increased in the sample collected after flowering (12.89%) than that collected before flowering (2.07%) [27]. Furthermore, the compound selina-3,11-dien-6-α-ol showed a higher percentage before flowering (9.24%), while the percentage of the cedryl acetate compound increased after flowering (8.10%). Caryophyllene oxide decreases after flowering (9.65 to 7.11%). Similarly, the levels of β-biotol and salvial-4(14)-en-1-one were elevated only during the first harvest period (5.16% and 4.66%, respectively). However, the percentages of eudesma-4- (15), 7-dien-1-ol and -hemachalol became elevated during the second period (5.85% and 5.67%, respectively) [27]. Similarly, other studies conducted in Morocco found spathulenol to be the most abundant compound in the EOs from *A. pyrethrum* [8,29]. According to these results and regardless of the time of harvest, the EOs from A. *pyrethrum* of Morocco can be classified as a spathulenol chemotype.

Moreover, in Algeria, the EOs from *A. pyrethrum,* extracted before and during flowering, have a germacrene-D chemotype (13.4 and 5.1%). While spathulenol maintains almost constant rates before and during flowering (4.7 and 4.2%, respectively) [30]. In fact, the chemical profile of the essential oils showed quantitative and qualitative changes, and this marked difference can be explained by several factors such as the harvest period, the biosynthesis process of these key components, the degree of plant maturity, geographical origin and genetics [31].

### 3.2. Non-Volatile Compounds

The chemical component in *A. pyrethrum* has been the subject of various studies. The phytochemical examination of the leaves, flowers, roots and flower heads revealed the presence of alkaloids, reducing the compounds and catechins tannins. In addition, this species contains other chemicals such as gallic tannins, sterols, triterpenes, mucilage, coumarins, geese and holocides [27,32] as well as trace minerals such as Fe, Zn, Cr, Cu, Cd, Pb, and Ni [33]. The highest content of flavonoids and polyphenols is observed in the flowers compared to the leaves and roots. The aerial parts are rich in tannins, while the alkaloids are very abundant in the roots.The most important compounds detected in the roots are the bioactive compounds *n*-alkylamides [34] and sharp brown resin, trace tannic acid, inulin, gum, various salts, anacycline, phenylethylamine, polyacetylenic amides I-IV, sesamin and lignin [32,35]. In a detailed study cited in [36], on the crystalline components of the roots, the authors have discovered a mixture of isobutyl amides of unsaturated acids with decadine as an important component. Similarly, the hydrogenation and acid hydrolysis of the radicals yield a mixture of decanoic, dodecanoic and tetradecanoic acids, which can be separated by reverse phase chromatography [37]. In addition, the pyrethrins found in the roots consist of a natural group of six chemically related esters: three (pyrethrin I, cinerin I and jasmolin I) are esters of chrysanthemum acid, and three (pyrethrin II, cinerin II and jasmolin II) are pyrethrins. The alcohol fractions detected in the study are pyrethrelone, cinerolone and jasmolone in pyrethrins 1 and 2, cinerin 1 and 2 and jasmolin 1 and 2, respectively [38]. The presence of pyrethrins and pellitorin in the roots of *A. pyrethrum* was confirmed in several studies (Figure 3) [27,32]. The isolation of pellitorin gives a new crystalline substance that crystallizes from chloroform-benzene in the white needles and is sparingly soluble in benzene with a melting point of 121 °C [37].

The study of ethanolic extracts from the roots, leaves and stems [33], the aqueous and methanolic extracts [32], the ethanolic and aqueous extracts [27] and the aqueous and methanolic extracts from the exposed roots [39] have reported the presence of alkaloids, flavonoids, tannins, steroids, triterpenes, reducing sugars, oils, saponins, anthraquinones and amino acids. The analysis through high-performance liquid chromatography with a UV detection coupled with an electrostatic spray ionization tandem mass spectrometry (HPLC/UV/ESI-MS) of the ethanolic extract of the dry roots, containing *n*-alkylamides, revealed the detection of 13 compounds [36,40]. Among these, six are considered new compounds: undeca-2*E*,4*E*-diene-8,10-dinoic acid *n*-methyl isobutylamide, undeca-2*E*,4*E*-diene-8,10-diene-8,10- diynoic acid isobutylamide, tetradeca-2*E*,4*E*-diene-8,10-diynoic acid tyramide, tetradeca-2*E*,4*E*,*XE*/*Z*-trienoic acid tyramide, tetradeca-2*E*,4*E*,*XE*/*Z*,*YE*/*Z*-tetraenoic isobutylamide and deca-2*E*,4*E*-dienoic acid-*n*-methyl isobutylamide [40].

The analysis through high-performance thin-layer chromatography (HPTLC) of the alcoholic extract of the roots shows the presence of picric acid, HNO_3_, CH_3_COOH, H_2_SO_4_, HCl, ferric chloride, aqueous KOH, alcoholic KOH, ammonia solution and iodine solution [16]. While other compounds were identified by GC-MS of the ethanolic extract from the same part of the plant as palmitic acid, 9,12-octadecadienoic acid (*Z*,*Z*)-, naphthalene, decahydro-1,1-dimethyl-, 7-tetradecenal, (*Z*)-, *n*-isobutyl-tetradeca-2,4-dynamite, benzofuran 2-carboxaldehyde and gamma-sitosterol [41].

The roots of *A. pyrethrum* also contain seven pure alkamides identified by MS and NMR. These compounds are deca-2*E*,4*E*,9-trienoic acid isobutylamide, deca-2*E*,4*E*-dienoic acid isobutylamide (pellitorin), deca-2*E*,4*E*-dienoic acid 2-phenylethylamide, tetradeca-2*E*, 4*E*-diene-8,10-dienoic acid isobutylamide (anacycline), undeca-2*E*,4*E*-diene-8,10-diynoic acid isopentylamine, dodeca-2*E*,4*E*-diene-4-hydroxy-2-phenylethyl amideacid and tetradeca-2*E*,4*E*,12*Z*-triene-8,10-dienoic acid isobutylamide [42,43]. In addition, the mixtures of two other alkamides were detected through column chromatography followed by HPLC as deca-2*E*,4*E*-dienoic acid 4-hydroxy-2-phenylethylamide and undeca-2*E*,4*E*-dien-8,10-dienoic acid 2-phenylethylamide [43].

A study on the ethanolic aqueous extracts of the flower heads, leaves and seeds of *A. pyrethrum* confirmed the presence of twenty compounds detected by GC-MS, among them sarcosine, *n*-(trifluoroacetyl)-butyl ester, levulinic acid, propanedioic acid, palmitic acid, morphinan 6-1, 4.5 alpha-epoxy-3-hydroxy-17-methyl, 2,4-undecadine-8,10-diene-*n*-tyramide and isovaleric acid. These compounds were first discovered in addition to the specific compounds such as alkylamides ((2,4)-*n*-isobutyl-2, 4-undecadien-8,10-diynamide, *n*-isobutyl-dodeca-2,4,8,10-tetraenamide, *n*-isobutyl-2,4-octadiene-6-monoynamide, *n*-isobutyl-2,4-heptadiene-6-monoynamide and *n*-isobutyl-2,6,8-datrieneamide) [5].

Another qualitative and quantitative study of the *A. pyrethrum* methanolic extract, focused on *n*-alkylamides, because they show distinct fractionation pathways that can be used as a basis for the component identification in this plant. Given this approach, twenty-one compounds have been identified, including twenty *n*-alkylamide and one organic acid. Two components were first observed in this plant: (2*E*,7*Z*)-*n*-isobutyl-2,7-tridecadiene-10,12-diynamide and (2*E*,6*Z*,8*E*)-*n*-isobutyl-2,6,8-diatrinamide [44]. In addition, new compounds were identified through quadrupole ultra-high performance liquid chromatography-time-of-flight mass spectrometry (UPLC-Q-TOF-MS): dodeca-2*E*,4*E*,*E*- trienoic acid-4-hydroxy phenylethylamide, tetradic acid–2*E*-deni-8,10-denoic acid, undeca-2*E*,4*E*-diene-8,10-diynoic acid 4-oh-phenylethylamide, tetradeca-2*E*,4*E*-ne-trienoic-8,10-diynoicacid and tetradeca-2*E*,4*E*-ne-trienoic acid [44].

Recently, two varieties of the *A. pyrethrum* species were studied [6]. The GC-MS analysis after the silylation of the extracts from different parts (roots, seeds, leaves and flower heads) of *A. pyrethrum* var. *pyrethrum* (L.) and A. *pyrethrum* var. *depressus* (Ball) Maire asserted the existence of twenty-two compounds. The results of this analysis also showed that the two varieties share the same compounds except for the compounds of *n*-isobutyl-2,4-heptadiene-6-monoynamide and cinnamic acid, which were detected in *A. pyrethrum* var. *pyrethrum* only. While the thiazolo [5,4-d] pyrimidine-7-amine and *n*-isobutyl-2,4-undecadien-8,10-dinamide compounds were restricted to A. *pyrethrum* var. *depressus* (Ball) only. In fact, the chemical analysis of the extracts of *A. pyrethrum* var. *pyrethrum* revealed the presence of the following components: *n*-isobutyl-dodeca-2,4,8,10-tetra enamide, *n*-isobutyl-2,4-octadiene-6-monoynamide, levulinic acid malonic acid palmitic acid morphinan-6-one, 4,5-epoxy-3-hydroxy-17-methyl, 2,4-undecadine-8,10-diene-*n*-tyramide, dodeca 2*E*, 4*E*, ne-trienoic acid and 4-hydroxyphenylethylamine, in all studied parts (roots, seeds, leaves and flower heads), while the cultivar A. *pyrethrum* var. *depressus* was characterized by the presence of four compounds: *n*-isobutyl-dodeca-2,4,8,10-tetra enamide, levulinic acid, palmitic acid and 2,4-undecadin-8,10-diene-*n*-tyramide [6].

## 4. Biological and Pharmacological Actions

*A. pyrethrum* is now a popular research plant, and has been studied for its pharmacological effects. Some of the health benefits are described below.

### 4.1. Antioxidant Activity

The search for natural antioxidants has been the subject of several studies on *A. pyrethrum* (Table 2) [6,8,39,45,46]. In a previous study, the author and his collaborators studied the antioxidant potency of the MeOH ext., the Aqu. ext. and the Chl. ext. from the stems and leaves of *A. pyrethrum* harvested in Algeria, using the DPPH˙ and FRAP methods [46]. Through the DPPH˙ method, they found that the MeOH extract in the species presents the most significant antioxidant potency (IC_50_ of 0.056 mg/mL), followed by the Aqu. ext. for IC_50_ of 0.114 mg/mL, and the Chl. ext. with IC_50_ of 0.154 mg/mL. Through the FRAP method, the results showed that the MeOH ext. has a stronger reducing power compared to the other extracts [46].

In 2017, Manouze et al. [39] determined the antioxidant potency of the MeOH ext. and the Aqu. ext. of the *A. pyrethrum* roots harvested in the Marrakesh-Morocco region. The authors used three methods: DPPH˙, FRAP, and BCB. For these three methods, the MeOH ext. recorded a significant antioxidant effect with IC_50_ 12.38, 50.89 and 107.07 μg/mL, respectively; the Aqu. ext. recorded an antioxidant effect with IC_50_ of 13.41, 60.17 and 120.66 μg/mL, respectively. In addition, Elazzouzi et al. [45] studied the antioxidant efficacy of the *A. pyrethrum* root extracts from the Timatidite region, Morocco. They reported that the MeOH ext., BuOH fra., AcEth fra. and Res pha. from the roots of *A. pyrethrum* had IC_50_ values of 0.152 mg/mL, 0.155 mg/mL and 0.144 mg/mL, respectively [45]. In 2020, the same author studied the antioxidant activity of the roots of *A. pyrethrum* using the DPPH method. The authors noticed that the IC_50_ of this substance is 30.50 mg/mL [8].

Recently, Jawhari and his collaborators studied the antioxidant power through DPPH and through FRAP of the MeOH ext. of the roots, leaves, flowers and seeds of two species of *A. pyrethrum* (A. *pyrethrum* var. *pyrethrum* (L.) and A. *pyrethrum* var. *depressus* (Ball) Maire) from the Timhdite region in Morocco. This study showed that the strongest antioxidant potency was observed in the leaves of A. *pyrethrum* var. *depressus* (Ball) Maire (IC_50_ = 0.03 mg/mL), which can be attributed to phenols, flavonoids and alkylamides of *A. pyrethrum* [6]. While the seeds of the same variety have a high reducing power (0.25 mg/mL). The roots of *A. pyrethrum* var. *pyrethrum* (L.) had the highest antioxidant activity, with a content of 708.74 mg ascorbic acid equivalent/g.

### 4.2. Antidiabetic Activity

Since antiquity, many diabetic subjects have used the traditional herbal remedies, in a range of formulations, as complementary therapies to control diabetic complications [47,48,49]. Among the numerous species cited in a catalogue previously published the *Pyrethrum* is described as a drug for the traditional therapy of diabetes, and this has been reported in the literature [42,47,49,50]. In the present literature review, an in vivo study by Shahraki et al. [51] revealed that the administration of the *A. pyrethrum* root alcoholic extract (ethylic alcohol (96%)) improved the diabetic impaired tissues in diabetic rats (100 and 150 mg/kg).

Some in vivo studies on diabetic rats [52,53], induced by alloxan or streptozotocin, respectively, have suggested the antihyperglycemic effect of an aqueous root extract of A. *pyrethrum* at a concentration of 300 mg/kg and 250 mg/kg, respectively. The elevated levels of the blood glucose in the diabetic rats returned to nearly normal levels after treatment. The species for a proposed study (*A. pyrethrum*) was subjected to an α-amylase inhibitory assay. The root ethanolic extract of *A. pyrethrum* significantly has antidiabetic properties in vitro, by inhibiting the dose dependent α-amylase activity (IC_50_ = 29.25 μg/mL) [42]. Moreover, this study supports that *A. pyrethrum* could be useful in the management of diabetes and previous results suggested that *A. pyrethrum* contains some compounds identified in the extracts as alkylamides, alkaloids and phenolics constituents [5,53]. Thus, the antidiabetic activity of these extracts in vivo needs to be evaluated prior to clinical use [54].

### 4.3. Anesthetic Activity

Local anesthetics are drugs that cause a loss of sensation that can be reversed when applied topically. This study aims to report the local anesthetic activity of the *A. pyrethrum* plant. The anesthetic activity of ethanolic, petroleum ether and aqueous extract from the *A. pyrethrum* through a Soxhlet apparatus, was tested using guinea pigs selected for an anterior study [55]. The results suggest that the ethanol extract in 1% and 2% concentrations show a significant local anesthetic activity. Similarly, 2% petroleum ether extract was more effective, followed by 1% petroleum ether and 2% aqueous extract. While the 1% aqueous reaction did not appear. Whereas, the histopathological analysis showed no adverse inflammatory changes in all extracts when compared with the normal tissue sample.

The aqueous and alcoholic extracts (2%) from the *A. pyrethrum* roots have shown that in experimental animals (frogs, guinea pigs and rabbits) have a longer duration as local anesthetics than xylocaine. This is the first primary study on the local anesthetic activity of this plant [56]. In a clinical assay, dental patients were subjected to an anesthetic study following oral surgery by comparing the activity of an alcoholic extract from the roots of *A. pyrethrum* (2% alcohol extract) with xylocaine. The plant extract was useful and safe (less than 2% concentration) and showed no side effects and had a prolonged anesthetic effect compared with xylocaine [57].

### 4.4. Insecticidal Activity

The insecticidal potential of *A. pyrethrum* has been confirmed for many years. This plant can eliminate several types of insects, such as beetles, whiteflies, candles, thrips, aphids, Mediterranean flour mites, leafhoppers, ants (except fire ants), aphids, crickets, fungal mosquitoes, pink slugs, cabbage worms, cocoon worms, Indian mealworms, mealybugs, pink beetles, spiders, trogoderma, etc. [38]. A study conducted on the hair of patients suffering from head lice demonstrated the efficacy of natural synergistic pyrethrins (presented in the form of an aerosol foam). This formulation is more effective compared to another, based on permethrin (in cream form). Likewise, the pyrethrin synergistic foam is effective as both an oviduct in a single application. This study also confirmed the high level of in vitro efficacy of oocytes and insecticides because the pyrethrin foam can eliminate an infestation with a single treatment without relying on residual insecticidal activity [58].

The alkaloids extracted from the roots of *A. pyrethrum* can be considered as biocidal molecules because they show a significant insecticidal effect on *Callosobrocus maculatus* adults. They significantly reduced the survival of bruchids reared on chickpeas (the coefficients of variation evolve from 23.04% to 82.22% in females and from 21.34% to 78.25% in males). The extracts also reduced fertility, which varies from 140.67% to 90.33%, fecundity (from 89.15% to 34.06%) and the success rate (from 83.83% to 13.82%) as the concentration increases. The number of infertile eggs of these insects increased from 15% to 59%, especially for the high concentrations [59]. Indeed, the obtained results show that the alkaloids constitute the most important chemical family in the roots of *Anacyclus pyrethrum*; they are extracted and analyzed through high performance liquid chromatography coupled with mass spectrometry (HPLC-MS/MS) to prove the presence of a major alkaloid *n*-isobutylamide, the pellitorine accompanied with a mixture of other alkylamides (ancycline, *n*-isobutyl-2,4-heptadiene-6-monoynamide, *n*-isobutyl-2,4-octadiene-6-monoynamide, (2,4)-dodecadiene-*n*-tyamide, *n*-isobutyl-2,4-hexadiynamide, acetanilide, *n*-methyl-isobutyl-2,4-decadienamide, etc.), which can be considered as molecules in promising biocidal properties.

The insecticidal potential of the essential oils of *A. pyrethrum* was first evaluated by Mokhtari et al. [7] with the goal of controlling *C. pipiens* larvae that cause a worldwide public health problem. The analysis of the results of larval killing tests for this oil showed that the total mortality rate (100%) of *C. pipiens* larvae was observed at a dose of 40 μL/mL. The EOs extracted from *A. pyrethrum,* containing spathulenol, germacrene-D and caryophyllene oxide as major compounds, might be used as a natural insecticide [7]. The root extract from *A. pyrethrum,* killed 90% of third instar larvae of *A. aegypti* at a concentration of 0.2 mg/mL, relative to the permethrin standard. It also showed a pesticide activity against *A. aegypti* eggs by 21.6% [59,60]. In fact, chemical composition of the root of *A. pyrethrum* showed the presence of n-hexadecanoic acid (palmitic acid) and 9-12-octadecadienoic acid (linoleic acid), which was reported to had larvicidal activities against *Aedes aegypti* [60].

The specific mechanism of action of *A. pyrethrum* is to disrupt the nervous system of insects. Furthermore, pyrethrins are toxic because they act on neurons causing paralysis and death, as reported in the studies [38].

### 4.5. Antidepressant Activity

It is estimated that the adult population suffers from episodes of depression during their lifetime [61]. Various plant extracts have been analyzed as potential antidepressant agents with validated models for testing antidepressant-like effects in animals, although other complementary studies have been used [35].

In 2010, a study evaluated the antidepressant activity of *A. pyrethrum* by various methods, such as locomotor activity, haloperidol-induced stimulation, forced swim test (FST), tail suspension test (TST), clonidine-induced hypothermia, hypothermia induced by hypothermia, clonidine-induced hypothermia and reserpine-induced hypothermia in Swiss male albino rats [62]. They found that the aqueous-alcoholic root extract of *A. Pyrethrum* showed an increase in the ambulatory behavior, suggesting a stimulating effect of the photometer. The extract from the *A. pyrethrum* root produces an antidepressant effect in the forced swim test and the tail suspension test, as it reduces immobility. The extract of the *A. pyrethrum* root was effective in reversing the hypothermia induced by clonidine and reserpine in rats at the doses of 100 and 200 mg/kg [62]. In another study conducted in 2011, the authors found that a forced swim test showed that the ethanolic extract of *A. pyrethrum* apparently acted as an antidepressant in mice. The decrease in motility was similar to the effects observed after the administration of a reference antidepressant, imipramine and an alleged involvement of the catecholamines in the antidepressant effect of the *A. pyrethrum* extracts could be suggested [63].

### 4.6. Antimicrobial Activity

The study of the plants, particularly the medicinal plants, constitutes a good avenue of exploration for the development of natural antimicrobials capable of inhibiting and/or killing microbes [64]. In several studies, the antimicrobial effect of the essential oil and the extracts of *A. pyrethrum* against microbial strains has been reported in Table 3.

In research, the susceptibility of the strains of *E. coli* (susceptible), *E. coli* (resistant), *S. aureus*, *P. aeroginosa* (susceptible), *P. aeroginosa* (resistant) and *K. pneumonia*, have been assessed vis-a-vis the EOs, different extracts and alkaloids from the roots of *A. pyrethrum*. The results showed that Aqu. mac. ext. characterized a significant activity against the sensitive *E. coli*, the resistant *E. coli* and *S. aureus* with inhibition diameters of 9 mm [27]. Another study conducted on *A. pyrethrum* harvested in Algeria showed an antimicrobial potency of the MeOH ext., the Chl. ext. and the Aqu. ext. for the stems and leaves tested against *L. monocytogenes*, *S. aureus*, *B. cereus* and *C. albicans*. Moreover, the decoctions, infusions and the MeOH ext. from the roots of *A. pyrethrum* showed an interesting antibacterial effect [65]. Furthermore, other results have shown that the MeOH exttract of *A. perithrum* has an inhibitory effect on *Escherichia coli* at the concentrations from 300 to 1000 mg/mL [66]. The MIC for this extract is 800 mg/mL, while the MBC for the MeOH ext. is 900 mg/mL [66]. The inhibitory effect of the Aqu. ext. for the *A. pyrethrum* roots against *C. albicans* was also shown by Chavan [67].

**Table 3 plants-11-02578-t003:** Antimicrobial activity of *Anacyclus pyrethrum* extracts.

Used Part	Extracts	Method	Tested Strains	Results (Inhibition Diameter in mm)	References
Roots	EO	Method of disk diffusion	*S. aureus*	7	[27]
	Aqu. mac. ext.		*E. coli* sensible	9	
			*E. coli* resistant	9	
			*S. aureus*	9	
			*P. aeruginosa* sensible	6	
			*P. aeruginosa* resistant	7	
			*K. pneumoniae*	7	
	MeOH ext.		*E. coli* resistant	7	
			*S. aureus*	7	
			*P. aeruginosa* resistant	7	
	Aqu. ext.		*E. coli* sensible	8	
Roots	MeOH ext.		*E. coli*	23	[66]
Stems/Leaves	EtOH ext.	Disc-diffusionmethod	*L. monocytogenes*	14	[46]
			*S. aureus*	20	
			*Bacilluscereus*	16	
			*C. albicans*	18	
	Aqu. ext.		*L. monocytogenes*	10	
			*S. aureus*	16	
			*Bacilluscereus*	12	
			*C. albicans*	15	
	Chl. ext.		*L. monocytogenes*	8	
			*S. aureus*	11	
			*E. coli*	10	
			*C. albicans*	10	
Leaves	Aqu. ext.	Disc-diffusion method	*C. albicans*	6 7 8	[67]

### 4.7. Anti-Inflammatory Activity

The inflammation is a manifestation of several diseases. Studies have shown an anti-inflammatory effect of various extracts of *A. pyrethrum* in a model of inflammatory edema in rats. Manouze and collaborators have studied, in vivo, the anti-inflammatory effect of the aqueous and methanolic extract of the *A. pyrethrum* roots on xylene-induced rat ear edema, and Freund-induced whole rat claw edema. These authors have found that the tested extracts significantly reduced the CFA-induced foot edema and the xylene-induced ear edema. The oral administration of 250 and 500 mg/kg of these extracts reduced the CFA-induced mechanical hypersensitivity reactions. This reduction started from 1 h and 30 min after the treatment and lasted up to 7 h. The chronic treatment of both extracts reduced the mechanical hypersensitivity in the persistent pain states induced by the CFA [37]. In addition, other results have shown that the aqueous-alcoholic extracts of the leaves, seeds, roots and flower heads of *A. pyrethrum* have highly potent anti-inflammatory activities on edema in rats. Following one hour of treatment, the inhibition ranged from 61% to 71% in the oral treatment groups [5]. While the percentage of the inhibition was greater in the percutaneous treated groups and ranged from 60% to 82%. At the fifth hour, the inhibition ratio for all samples increases. Moreover, the flavonoids have a membrane-stabilizing effect by reducing the vasodilatation, which ameliorates the strength and integrity of the blood vessel walls, while alkaloids (pellitorin), including alkylamides, may act through the prevention of neurogenic inflammation [5,6]. In addition, in another work, various extracts (Aqu. ext., MeOH ext., and Chl. ext.) from *A. pyrethrum* showed a significant anti-inflammatory activity [68].

### 4.8. Aphrodisiac Activity

Throughout the ages, man has constantly sought to develop, maintain or aggravate his sexual ability, or to stimulate sexual desire. One of the most common methods is the use of the aromatic and medicinal plants. Among these plants, *A. pyrethrum* has been used for centuries for its therapeutic virtues, especially for aphrodisiac purposes. A study showed that the aqueous extract of the roots of *A. pyrethrum* increases body weight and reproductive organs, and thus increases the number of epididymal spermatozoa in rats. The aqueous extract also reduced the abnormal sperm count in rats [69]. In addition, the results of a study by Sharma et al. [68] on the MeOH ext. of the *A. pyrethrum* roots showed that this extract has an androgenic potential and can improve male fertility by promoting spermatogenesis [70]. The queous extract from the roots of *A. pyrethrum* increases sperm count and fructose levels in the seminal vesicle. It also improves sexual behavior in male rats. The extract had a dose-dependent effect on sperm count and sperm fructose concentration, which was significantly increased [71].

### 4.9. Anticonvulsant Activity

The anticonvulsant activity of *A. pyrethrum* has been extensively studied and proven by numerous studies. A recent study showed that the ethanolic aqueous extract of *A. pyrethrum* at a dose of 500 mg/kg increased seizure latency time in the first experimental group, compared to the first positive control [72]. Thus, significant changes were observed in the mean seizure score for the first and second positive control groups (*p* < 0.001) and for the first and second positive control groups (*p* < 0.05). In addition, the aforementioned study concluded that the blocked GABAergic A and B receptors (cervical proteins) are involved in epileptic seizures [72].

Furthermore, in other study, it was shown that the petroleum ether extract of *A. pyrethrum* can provide a significant protection against seizures induced by pentylenetetrazole (PTZ) [73]. In addition, this dose-dependent extract increased the latency of induced seizures and reduced the duration of epilepsy. Interestingly, the extracts showed a reduction in neurological symptoms and seizure severity compared to the negative control [73]. In the anticonvulsant activity of the *A. pyrethrum* roots evaluated using an experimental model of pilocarpine-induced epilepsy in mice [74]. The possible anticonvulsant mechanism was studied by testing the effect of atropine (2 mL/kg), scopolamine (1 mg/kg) and seizure severity, latency time, total seizure duration and protection against mortality, which were recorded. The ethanol extract and alkylamides increased the seizure onset time and decreased the seizure duration compared to the control group (*p* < 0.001). The protection against seizures was complete. The co-administration of the ethanolic extract of *A. pyrethrum* and alkylamides with atropine completely abolished the seizures induced by pilocarpine. These observed effects could be related to the effect of the extract and the alkylamides of *A. pyrethrum* on cholinergic receptors, and significantly, the same effects have been reported with the use of cholinergic receptor antagonists [74].

Regarding the study provided by Kalam et al. [75], two experimental models of epilepsy, the PTZ-induced seizure test and the extreme electric shock test, were performed. It was found that the aqueous ethanolic extract of *A. pyrethrum* significantly (*p* < 0.05) reduced the posterior tibial extensor tonic phase in maximal electroshock-induced epilepsy. In PTZ-induced seizures, the tested extract delayed the onset of the first seizure, tonic and clonic seizures, and a significant decrease in the total number of seizures and the duration of tonic and clonic seizures (*p* < 0.05). The ethanolic aqueous extract of *A. pyrethrum* was used at the highest dose that protected all animals from death, while the percentage of protection from death at the lowest doses was 33% [75].

In another study, the effect of the oral administration of an aqueous alcoholic extract of the *A. pyrethrum* root (100, 250 and 500 mg/kg) on PTZ-induced maturation, spatial memory, oxidative stress and rho kinase (ROCK II) was evaluated in albino mice [76]. Pre-treatment with the extract at doses of 250 and 500 mg/kg showed a significant increase in myoclonic latency and a delay in the development of myoclonus. A significant reduction in the mortality was observed at higher doses of the extract (250 and 500 mg/kg). Pretreatment with the root extract significantly increased the number of platform crossings and reduced the escape latency, in contrast to the PTZ group, showing protection against memory deficits. The PTZ-induced induction increased the ROCK II expression; while the pretreatment with this extract attenuated the increase in the ROCK II expression [76]. Moreover, the anticonvulsant potential of the ethanolic extract of the *A. pyrethrum* roots was shown to be evaluated at doses of 100–800 mg/kg *versus* PTZ, bicuculline (BCL) and electroconvulsive current augmentation (ECS) [77]. The ethanolic extract obtained by soaking revealed a significant anticonvulsant potential (*p* < 0.001) against PTZ (70 mg/kg) in a dose-dependent manner, but against BCL (30 mg/kg) this effect was only detected at a dose-dependent manner 800 mg/kg (*p* < 0.001). However, this extract did not protect the mice from convulsions caused by the increased current electric shock (*p* > 0.05). This study hypothesized that the anticonvulsant effect of the ethanolic extract of the *A. pyrethrum* roots was likely mediated by enhancing the GABAergic neurotransmission. The same group of researchers found, in 2009, that the chloroform fraction of the roots of *A. pyrethrum* is able to significantly delay the onset of seizures induced by pentylenetetrazole (*p* < 0.001). Thus, it was able to significantly increase the seizure threshold in the current increased electroshock test in Swiss albino mice [77].

### 4.10. Other Activities

Other studies showed that the extracts of *A. pyrethrum* have other pharmacological activities, such as hepatoprotective [78], anticancer [79], neuropharmacology [63], immunostimulants and immunomodulators [80,81,82].

A study founded that the aqueous-alcoholic extract of the *A. pyrethrum* root had a hepatoprotective effect against isoniazid and rifampicin-induced hepatotoxicity in rats [78]. Moreover, *A. pyrethrum* can be used as a new therapeutic product for treating colorectal cancer. Another work showed that the *A. pyrethrum* extract significantly inhibited the growth of cancer cells (human colorectal carcinoma (HCT)) and that this extract could successfully induce apoptosis in HCT cells [79].

In 2011, authors demonstrated that the ethanolic extract of *A. pyrethrum* has significant neuropharmacological activities [63]. The in vivo immunostimulating effect for *A. pyrethrum* has been investigated [80], there was a better stimulation index at a dose of 50 μg/mL than that obtained with the in vitro effect (50 mg/mL) [80]. In fact, the root extracts of *A. pyrethrum* exhibited a better immunomodulatory activity. The aqueous extract, administered orally to rats, showed a higher activity at a dose of 10 mg/kg [80]. Similarly, the methanolic extract, tested on rats, has an immunomdulatory effect at 200 mg/kg [81]. while the petroleum ether extract of *A. pyrethrum*, tested at 50 and 100 mg/kg doses, was able to overcome the cyclophosphamide-induced immunosuppression [82].

## 5. Toxicological Evidence

Currently, in Morocco as elsewhere, many plants with medicinal properties cause poisoning, and therefore constitute a rather serious public health problem [83]. In addition, a plant is considered toxic when it contains one or more substance harmful to humans or animals and its use causes various more or less serious disorders. This intensity depends on many factors, on the one hand, the portion consumed, dose, if fasting or not, and on the other hand, the age of the user and the conditions concerning the consumption of the plant [48,83]. Asteraceae are among the families most widely used in herbal medicine in most Mediterranean countries, as they are the richest in epiphytes [45,83], and they are also a reservoir of poisonous plants. Morover, several surveys revealed that all parts of *A. pyrethrum* were declared toxic by all respondents [84], and their use should be systematic, controlled and a safe use established. Furthermore, pharmacological studies and toxicological testing must be conducted to translate the knowledge using traditional plants into scientific knowledge [48,83,85,86]. This is how, on the basis of this information, we have decided to investigate the toxicity of the *A. pyrethrum* species.

In fact, since studies of the phytochemical analysis of *A. pyrethrum* have confirmed the presence of interesting compounds, it will not be without toxicity, and many accidents have been reported after therapeutic use. It thus causes toxic symptoms and can cause skin irritation to the mucous membranes as well as nausea, fainting and respiratory disturbances. Similarly, the oral administration can cause gastrointestinal irritation, gastroenteritis, colic, diarrhea, cramps and severe headaches [87]. Finally, through its fumes, it can lead to headaches, ringing in the ears and even fainting [88]. The toxic manifestation of *A. pyrethrum* is due to the presence of toxic agents, namely pyrethrins and unsaturated amides such as pyrethrin and anacycline. The latter in high doses cause discomfort, nausea and colic. Indeed, the isobutylamides act as nerve toxins by blocking the sodium channel. The development of isobutylamides as an agricultural product was terminated, as all analogues were unstable in the field [22,30,87,89,90,91,92].

Limited toxicity studies on the *A. pyrethrum* species are currently available in the literature. In the results of a previous study, the hydroethanolic extracts of the different parts of the *A. pyrethrum* (L.) varieties were not toxic at low concentrations [90]. Whereas some toxic effects were detected in mice treated at 2000 mg/kg. At this dose, the mice displayed histopathological changes in the liver, kidneys and spleen characterized by hepatic distress, inflammatory infiltration, focal tubular necrosis, vascular congestion and lymphoid hyperplasia [90]. A subchronic toxicity of the root ethanolic extract (1000 mg/kg), orally administered to rats, was evaluated. This study revealed that there were no mortalities or adverse effects. Furthermore, this extract has no treatment for concomitant toxic irregularities. Therefore, this study indicates that the ethanolic extract is safe for chronic treatments [93]. We mention that these results are in agreement with those reported in [39,94]. At a concentration of 5000 mg/kg of the aqueous methanolic and ethanolic root extracts, respectively, the results showed that there were no toxicity-related symptoms, mortality and weight changes in the body and organs were observed. This indicates that the administration of the crude extracts has a negligible level of toxicity in growing animals [42]. Furthermore, various other extracts of *A. pyrethrum* (petroleum ether, chloroform, ethyl acetate, acetone, ethanol, water extracts) have also been the subject of toxicological studies on mammals, and similar results are shown [35,85,95]. Therefore, although there is no toxicological study in humans, *A. pyrethrum* is highly effective against many types of insects, but its toxicity to warm-blooded animals is low, and no obvious harmful effect was found due to its rapid biotransformation [96,97].

A detailed study of *A. pyrethrum* demonstrates that the components of pyrethrins have a low to moderate acute toxicity via oral, dermal and inhaled human, and the chronic exposure includes neurobehavioral, thyroid and hepatic effects [98]. Therefore, another study reported that the minimum toxicity of this vertebrate neurotoxin was approximately LD_50_ of 1500 mg/kg [89]. Subsequently, this relatively low toxicity to mammals results in large part from factors that prevent its entry into the nervous system, such as the metabolic detoxification. The excessive exposure to some types of pyrethroids may cause skin irritation in sensitive individuals [99]. A recent study suggested that the assessment of toxicity is determined mainly in the ethyl acetate extract of the granulated roots, which showed LC_50_ = 249.3 μg/mL. This value is considered “toxic”, therefore, the fractionation assay of this active extract leads to the isolation of four alkamides, including pellitorin, and this effect may be associated with their presence [100].

## 6. Methodology

This review was conducted between May 2022 and September 2022 by exploring the global literature that directly relates to the botanical aspect, traditional properties, phytochemical contents, pharmacological action and poisonous properties of *A. pyrethrum*. All information about this plant was collected via libraries or electronic databases (PubMed, Elsevier, Web of Science, Scopus, Google Scholar and Springer). Various keywords were used during the bibliographic research including: *Anacyclus pyrethrum*, *Pyrethrum*, Asteraceae, phytochemistry, pharmacological activity and toxicity. The information was gathered and summarized in table form where appropriate.

## 7. Conclusions

*Anacyclus pyrethrum* is one of the interesting medicinal plants of Asteraceae family, endemic to Morocco and Algeria, it is used in traditional medicine to treat many diseases. In the present review, we have reported on the literature, botany, traditional use, phytochemistry, pharmacological use and toxicology of *A. pyrethrum*. The collected data confirm the pharmacological potential of this species. In fact, experimental studies have been studied for some pharmacological activities, such as antimicrobial, antioxidant, anti-inflammatory, hepatoprotective, anticancer, neuroprotective, antidepressant and aphrodisiac activities. In addition, numerous studies have shown the richness of this species in secondary metabolites; namely terpenoids, polyphenols, alkaloids, etc. These justifies its extensive traditional therapeutic use. However, it is worthwhile to mention that there are few studies on the fractionation of its compounds and on their isolation.

## Figures and Tables

**Figure 1 plants-11-02578-f001:**
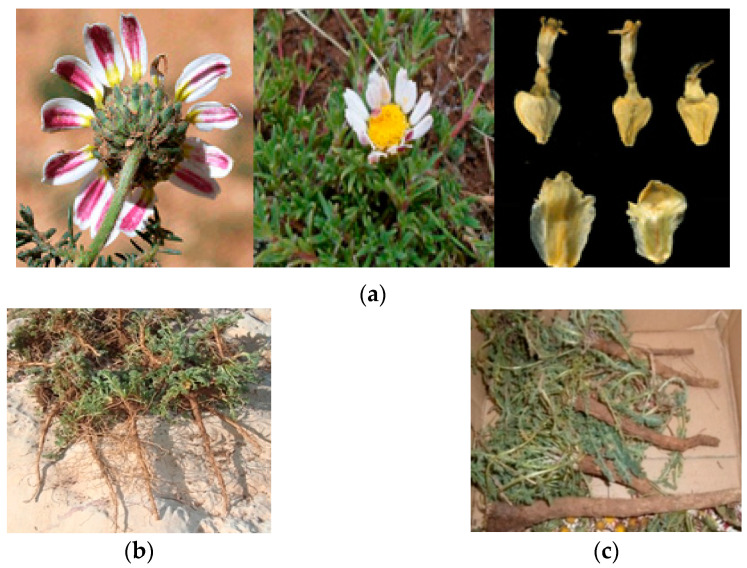
Morphological appearance: Flowers, ray florets and achene of *A. pyrethrum* (**a**), *A. pyrethrum* var. depressus ‘Tigendast’ roots (**b**) and *A. pyrethrum* var. pyrethrum ‘Igendas’ roots (**c**).

**Figure 2 plants-11-02578-f002:**
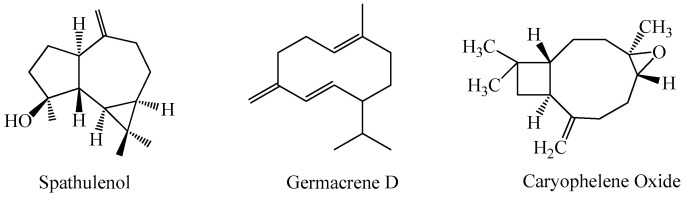
Chemical structures of the volatile components in *Anacyclus pyrethrum*.

**Figure 3 plants-11-02578-f003:**
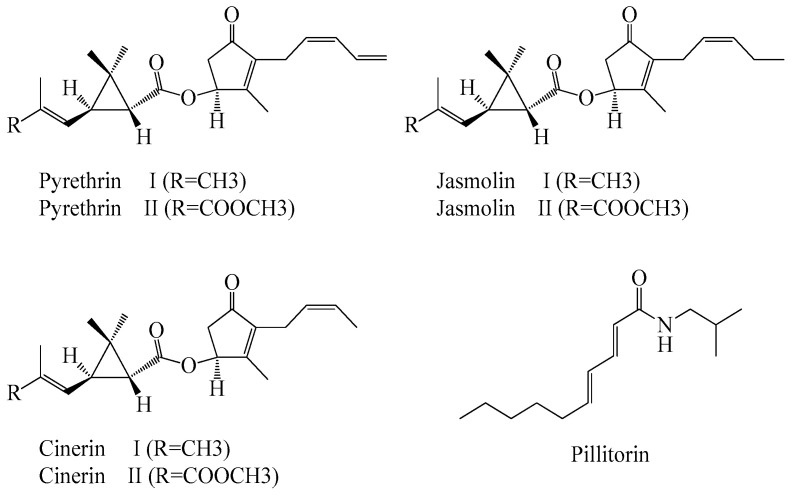
Chemical structures of the non-volatile components isolated from *Anacyclus pyrethrum*.

**Table 1 plants-11-02578-t001:** Therapeutic actions of *A. pyrethrum*.

Used Part	Mode of Preparation	Medicinal Use	References
Root	Decoction	Stomach diseases and stomatitis	[20]
Stem	Powder	Cysts of the reproductive system	[21]
Root	Powder	Rheumatic, gastrointestinal, oral diseases, respiratory, genitourinary, skin and dermatitis diseases	[22,23]
Root	Decoction/Powder	Osteoarthritis disorders, stomatitis, inflammation of the urinary and genital organs, and respiratory diseases	[24,25]
Root	Infusion/Decoction	sore throats, toothache and skin revitalization	[26]
Root	Powder/Decoction	Articular rheumatism, dental pain, intestinal pain and colic	[1]

**Table 2 plants-11-02578-t002:** Antioxidant activity of *A. pyrethrum* (var. *pyrethrum* (L.) and var. *depressus* (Ball) Maire).

Variety	Used Part	Extract	Method	Result (in IC_50_ or Absorbance (A))	References
var. *Pyrethrum*	Roots Flowers Seeds Leaves	EtOH	DPPH˙	0.18 mg/mL 0.16 mg/mL 0.01 mg/mL 0.04 mg/mL	[6]
	Roots Flowers Seeds Leaves		ABTS	0.14 mg/mL 0.07 mg/mL 0.05 mg/mL 0.03 mg/mL	
	Roots Flowers Seeds Leaves		FRAP	1.19 mg/mL 1.08 mg/mL 0.49 mg/mL 0.38 mg/mL	
var. *Depressus*	Roots Flowers Seeds Leaves		DPPH˙	0.07 mg/mL 0.08 mg/mL 0.04 mg/mL 0.03 mg/mL	
	Roots Flowers Seeds Leaves		ABTS	0.05 mg/mL 0.05 mg/mL 0.05 mg/mL 0.03 mg/mL	
	Roots Flowers Seeds Leaves		FRAP	0.38 mg/mL 0.59 mg/mL 0.25 mg/mL 0.23 mg/mL	
var. *Pyrethrum*	Stems/leaves	MeOH ext.	DPPH˙	0.056 mg/mL	[46]
		Aqu. ext.		0.114 mg/mL	
		Chl. ext.		0.154 mg/mL	
	Root	MeOH ext.	DPPH˙	12.38 µg/mL	[39]
			FRAP	50.89 µg/mL	
			BCB	107.07 µg/mL	
		Aqu. ext.	DPPH˙	13.41 µg/mL	
			FRAP	60.17 µg/mL	
			BCB	120.66 µg/mL	
		MeOH ext.	DPPH˙	0.15 mg/mL	[45]
		AcEth ext.		0.14 µg/mL	
		BuOH ext.		0.15 mg/mL	[8]
		HE	DPPH˙	30.50 mg/mL	

## Data Availability

Not applicable.

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
