# Peer review of "Phytochemistry, Biological and Pharmacological Activities of the Anacyclus pyrethrum (L.) Lag: A Systematic Review"

_plants, 2022, doi:10.3390/plants11192578_

Round 1

Reviewer 1 Report

Manuscript entitled: Phytochemistry, biological and pharmacological activities of  Anacyclus pyrethrum: A Systematic Review was an interesting read. The article generally reflects Anacyclus pyrethrum (Asteraceae) investigated properties, chemical composition, potential use of this plant. One of the main positive features, people who read this article will get a lot of information from one article, it will not be necessary to go through many articles to get different information. 101 literature reviews have been carried out with information published in recent years, which is positive, because with the development of science and research opportunities, a lot of useful information about medicinal plants, their composition, the effects of using them, etc. has been obtained using innovative research approaches in recent years. 

Some objections that should be corrected before accepting the work are the writing of significant numbers in the tables, for example, 0.18±0.005 mg/mL it is not correct it is need 0.18......and ±0.005 or in table 2 you write 7±0 (what does it mean?)

Author Response

Manuscript entitled: Phytochemistry, biological and pharmacological activities of Anacyclus pyrethrum: A Systematic Review was an interesting read. The article generally reflects Anacyclus pyrethrum (Asteraceae) investigated properties, chemical composition, potential use of this plant. One of the main positive features, people who read this article will get a lot of information from one article, it will not be necessary to go through many articles to get different information. 101 literature reviews have been carried out with information published in recent years, which is positive, because with the development of science and research opportunities, a lot of useful information about medicinal plants, their composition, the effects of using them, etc. has been obtained using innovative research approaches in recent years.

RESPONSE: Thank you so much for your kind comments about our manuscript.

Some objections that should be corrected before accepting the work are the writing of significant numbers in the tables, for example, 0.18±0.005 mg/mL it is not correct it is need 0.18......and ±0.005 or in table 2 you write 7±0 (what does it mean?)

RESPONSE: Thank you so much for comments. Now, these mistakes have been corrected.

Reviewer 2 Report

Thanks for the opportunity to review this research. The manuscript entitled „ Phytochemistry, biological and pharmacological activities of Anacyclus pyrethrum: A Systematic Review” have described the updated information on Anacyclus pyrethrum phytochemical and pharmacological properties. The subject of the manuscript is topical, but I recommend the publishing of the paper after the necessary corrections.

1.     There are several typographical mistakes as well in whole manuscript. Therefore, the author’s thoroughly careful check the language and typo mistake to minimize the error.

2.     Check and format the citations in the whole manuscript. Also, Appropriate references must be provided to explained the background, what is already done and why this study carried out. Hypothesis statement is missing in the introduction section.

3.     The manuscript provided by authors is difficult to follow and verify due missing critical details in the methodology section.

4.     The manuscript conclusion must be substantially rewritten.

Author Response

Thanks for the opportunity to review this research. The manuscript entitled „ Phytochemistry, biological and pharmacological activities of Anacyclus pyrethrum: A Systematic Review” have described the updated information on Anacyclus pyrethrum phytochemical and pharmacological properties. The subject of the manuscript is topical, but I recommend the publishing of the paper after the necessary corrections.

  1. There are several typographical mistakes as well in whole manuscript. Therefore, the author’s thoroughly careful check the language and typo mistake to minimize the error.

RESPONSE: We revised the manuscripts and all mistakes have been corrected.

  1. Check and format the citations in the whole manuscript. Also, Appropriate references must be provided to explained the background, what is already done and why this study carried out. Hypothesis statement is missing in the introduction section.

RESPONSE: We revised the manuscript according to the guide of authors and all mistakes have been corrected.

  1. The manuscript provided by authors is difficult to follow and verify due missing critical details in the methodology section.

RESPONSE: This section of the manuscript has been improved.

  1. Themanuscript conclusion must be substantially rewritten.

RESPONSE: The conclusions have been rewritten.

Reviewer 3 Report

This is an ethnopharmacological review on the beneficial properties of the plant Anacyclus pyretrhum (L.) Lag. Based on popular experiences in Morocco.

This type of work is always interesting since it highlights potential medicinal sources of plants, less used in European, American or Japanese pharmacopoeias.

However, even though the work was designed correctly, it is a very general study and needs corrections to be published.

In the title, put the full name of the plant: Anacyclus pyrethrum (L.) Lag.

Asteraceae is not written in italics.

Throughout the work the plant must be referred to as A. pyretrhum

In the botanical description, the external flowers are not petals, but ray florets.

Higher quality photographs with more detail of the plant would be needed.

I do not understand the male root and female root, since this plant is a hermaphrodite.

A world distribution map of this plant would be needed.

At no time is any molecule or group of molecules responsible for the supposed therapeutic actions named. Nor does a graph appear with the most representative molecular structures.

To follow point 4, it is important to put a table at the beginning with medicinal applications, even if they are discussed later in the text.

Finally, although the literature says otherwise, my experience tells me that Asteraceae, in general, are not very toxic.

Author Response

This is an ethnopharmacological review on the beneficial properties of the plant Anacyclus pyretrhum (L.) Lag. Based on popular experiences in Morocco. This type of work is always interesting since it highlights potential medicinal sources of plants, less used in European, American or Japanese pharmacopoeias. However, even though the work was designed correctly, it is a very general study and needs corrections to be published.

In the title, put the full name of the plant: Anacyclus pyrethrum (L.) Lag.

RESPONSE: Done

Asteraceae is not written in italics.

RESPONSE: Done

Throughout the work the plant must be referred to as A. pyretrhum

RESPONSE: Done

In the botanical description, the external flowers are not petals, but ray florets.

RESPONSE: Thank you so much for your comment. This mistake has been corrected.

Higher quality photographs with more detail of the plant would be needed.

RESPONSE: Done

I do not understand the male root and female root, since this plant is a hermaphrodite.

RESPONSE: Thank you so much for your comment. This mistake has been corrected.

A world distribution map of this plant would be needed.

RESPONSE: There is no world distribution map in the literature of this plant.

At no time is any molecule or group of molecules responsible for the supposed therapeutic actions named. Nor does a graph appear with the most representative molecular structures.

RESPONSE: Thank you so much for your comment. This information has been included in the manuscript

To follow point 4, it is important to put a table at the beginning with medicinal applications, even if they are discussed later in the text.

RESPONSE: Thank you so much for your comment. This information has been included in the manuscript

Finally, although the literature says otherwise, my experience tells me that Asteraceae, in general, are not very toxic.

RESPONSE: Thank you so much for your comment.

Round 2

Reviewer 3 Report

None

Author Response

Many thank for improving our manuscript with your suggestions